# Morphometric Relationships, Growth and Condition Factors of Critically Endangered Chinese Pangolin (*Manis pentadactyla*)

**DOI:** 10.3390/ani12070910

**Published:** 2022-04-02

**Authors:** Tulshi Laxmi Suwal, Meng-Jou Chi, Chi-Feng Tsai, Fang-Tse Chan, Kuei-Hsien Lin, Kurtis Jai-Chyi Pei

**Affiliations:** 1Department of Tropical Agriculture and International Cooperation, National Pingtung University of Science and Technology, Pingtung 91201, Taiwan; 2Small Mammals Conservation Research Foundation, Kathmandu 44600, Nepal; 3IUCN SSC Pangolin Specialist Group, C/o Zoological Society of London, Regent’s Park, London NW1 4NR, UK; 4Pingtung Rescue Center for Endangered Wild Animals, National Pingtung University of Science and Technology, Pingtung 91201, Taiwan; savvy.chi@gmail.com; 5Endemic Species Research Institute, 1 Minsheng East Road, Jiji Town, Nantou 226000, Taiwan; cftsai@tesri.gov.tw (C.-F.T.); cft01@tesri.gov.tw (F.-T.C.); sds0308@tesri.gov.tw (K.-H.L.); 6Institute of Wildlife Conservation, College of Veterinary Medicine, National Pingtung University of Science and Technology, Pingtung 91201, Taiwan

**Keywords:** bodyweight, Chinese pangolin, relative condition factor, standardized guideline, total length

## Abstract

**Simple Summary:**

The pangolin is the only scaly mammal in the world. Among the eight extant pangolin species, the Chinese pangolin (CP) is most threatened with extinction. Thus, this species is categorized as “Critically Endangered” by the International Union for Conservation of Nature (IUCN) and listed under “Appendix I” of the Convention on International Trade in Endangered Species of Wild Fauna and Flora (CITES). Taiwan is a single Formosa Island where the population of CP has been increasing for the last decade due to continuous conservation initiatives. The presence of sufficient samples of CP that are adapted to distinct environmental conditions could provide valid morphometric results. However, the morphometric relationships, growth type, and condition factors have not yet been enumerated for any pangolin species including Chinese pangolin. The study included 282 rescued and rehabilitated individuals from central and southern Taiwan. Adult male and female pangolins measure between 75.2–103 cm and 66–114.9 cm from the snout to the tip of the tail and weigh between 4–7.6 kg and 3–5.8 kg, respectively. Adults also showed negative allometric growth (b < 3) as their length increased. However, the average relative condition factor (KR) was 1.02 ± 0.16 and showed the majority of rescued adults in normal (59.7%) and good (39%) health conditions.

**Abstract:**

Morphometric relationships and condition factors are crucial to quickly understanding the fitness and well-being of animals. Total length (cm) and bodyweight (g) of 282 (male = 167 and female = 115) pangolins were accounted for in this study which was received and rehabilitated in Pingtung Rescued Center, Pingtung and Endemic Species Research Institute, Nantou, Taiwan. The allometric equation; W = aL^b^ was used to estimate the length-weight relationships where R^2^ = 0.70, a = 0.61, and b = 1.98. The ratio of total body length and tail length was between 1.1 and 2.7. Pangolins exhibited negative allometric growth (b < 3) as their length increased. The average relative condition factor (K_R_) was 1.04, indicating a state of good health for rescued individuals. However, Fulton condition factor (K_F_) and K_R_ fluctuated in different months but was significantly heavier during the wet seasons. This study recommended to release healthy animals with K_F_ of 0.8 or higher back into the wild in order to increase their survival rate. The study, thus will be helpful to promote standardized guidelines for conducting physical measurements and understanding health status. Additionally, it will support the recovery of this threatened species by husbandry and diet management in ex-situ and in-situ conservation.

## 1. Introduction

Chinese pangolin (*Manis pentadactyla*) is one of the most illegally traded mammals for meat consumption, traditional medicine, and souvenirs. Wildlife authorities are constantly confiscating and seizing hundreds of live pangolins from illegal traders each year around the world [1]. The illegally trafficked individuals are often found dehydrated, nutritionally stressed, and have injuries from when they were hunted [2]. The rescued live individuals are treated and rehabilitated in captivity so that they can be returned to their native habitats [3]. The Chinese pangolin occurs in South and Southeast Asia; Bangladesh, Bhutan, China, India, Lao PDR, Myanmar, Nepal, Taiwan, Thailand, and Vietnam with a distribution range overlapping with that of the Sunda pangolin (*Manis javanica*) [4] and Indian pangolin (*Manis crassicaudata*) [5]. This species utilizes a wide range of forests, including primary and secondary tropical and sub-tropical forests and grasslands, as well as agricultural fields [6]. Pangolins are rarely observed in the wild due to their increasing rarity, secretive, solitary, and nocturnal nature. Over-exploitation has resulted in the extirpation of the species from parts of its range countries and its current population trend is highly decreasing [1]. Globally, the Chinese pangolin is classified as Critically Endangered on the IUCN Red List of Threatened Species and listed under Appendix I of the Convention on International Trade in Endangered Species of Wild Fauna and Flora [7]. It is also protected by national laws in all range countries.

Pangolin is an idiosyncratic mammal with a typical morphological characteristic. The morphological parameters including length-weight relationships (LWRs), growth and condition factors (K) of Chinese pangolin are lacking [8,9]. Previously, only two studies on Chinese pangolin’s morphometric measurements have been carried out, namely [10] in Taiwan and [8] in China. Both studies, however, used very small sample sizes (maximum 20 samples) for the physical measurements. Additionally, there is not any information on LWRs and the condition factors of any species of pangolin to date. The LWR is not only a morphological characteristic of species, but it also provides information about general health, habitat, and body condition [11]. The Fulton condition factor (K_F_) and relative condition factor (K_R_) are key determinants of an individual animal’s provide overall fitness, and its implications regarding health in terms of biomass fitness of a population in a habitat [12,13].

The body condition of an animal refers to its energetic state, so an animal in good condition has more energy reserves (usually a fat body) than an animal in poor condition. In mammals, the amount of fat that an individual carries can have significant fitness consequences. For instance, individuals with larger fat reserves may have better fasting endurance and better survival than individuals with smaller reserves [14]. Condition factors (K) for pangolins are the first aid facility, which contributes to a quick assessment of pangolins’ physical health status [15]. Therefore, first-aid facilities in many countries can use the K value to evaluate individuals and allocate more resources to help individuals with higher K values. A higher K value might indicate a better chance of survival. Therefore, this study examined biometric morphometrics of rescued pangolins, with the aim to (i) (investigate morphometric relationships and growth conditions, (ii) examine condition factors to assess fitness and well-being in different age groups, and (iii) assess monthly and seasonal variations in LWRs and K in adults. In addition, this study recommends the K-value for quick decision-making regarding healthy and non-healthy individuals in the rescue centers and customs departments for further treatments, as well as other procedures, such as rehabilitation and release.

## 2. Materials and Methods

### Data Collection and Analysis

The information on rescued Chinese pangolins received in the Pingtung Rescue Center for Endangered Wild Animals (PTRC), Pingtung (n = 178), and the Endemic Species Research Institute (ESRI), Nantou (n = 104), Taiwan, was managed by excluding missing data and injured individuals. These rescued pangolins were in temporary care and released back into a safe natural habitat. Length and weight measurement was conducted during the first aid treatment following the wild mammals handling procedure [16] under the Wildlife Conservation Act of Taiwan. Bodyweight was measured accurately by a digital balance in grams (g). For the body length, the pangolin was first uncurled, straightened, and lying on its abdomen (with ventral/dorsal side up) on the table, and the length from the tip of the snout to the end of the tail region was measured by centimeter (cm) measuring tapes. Young Chinese pangolins are normally 80–180 g at birth [9]. Based on the bodyweight, the animals were categorized into different age classes in this article, such as both sexes are around 1 kg at 6 months old, 2 kg at 1 year old; females are around 3 kg at 2 years old, and males are around 4 kg at 2 years old; females are around 4 kg at 3 years old and level off with seasonal fluctuation, and males are around 5 kg at 3 years old and level off with seasonal fluctuation (N. C.-M. Sun, personal communication).

The allometric equation for length-weight relationships (LWRs), W= aL^b^, was used to estimate the relationship between the weight (W; g) of the pangolins with their total length (L; cm). Logarithmic transformation of LWRs parameters and estimation by using the linear regression of data transformed to base 10 logarithms; LogW = Loga + b LogL [17]. Based on the value of b, the growth pattern was classified into isometric growth when the ideal value of b was equal to 3. When b was less than 3, the individuals were slimmer with increasing length and the growth is negatively allometric. When b was greater than 3, the individual was heavier showing positive allometric growth and reflecting optimum conditions for the growth [18]. To establish LWRs with respect to variation [19], gender was grouped into male and female.

The Fulton condition factor (K_F_) and relative condition factor (K_R_) were developed for quick evaluating the physical status or well-being of the confiscated or rescued pangolins, based on the assumption that heavier pangolins of a given length are in better conditions. The factor is calculated using the Fulton 1904 suggested formula for the Condition Factor (K_F_) = weight/length^3^ × 100 [20] When weight is measured in g and length in cm. The relative condition factor (K_R_) was calculated by the ratio of observed weight (Wo) and calculated weight (Wc). K_R_ = 1 or near to 1 indicates a “normal” health condition whereas K_R_ greater than 1 (K_R_ > 1) is good and fat, however, less than 1(K_R_ < 1) indicates poor health status with a skinny body [17]. To establish body condition with respect to seasonal variation, the season was divided into two seasons as; Wet Season is May-October and Dry Season is November-April [2,21].

All the data were managed in Excel and analyzed using IBM SPSS statistics 26. Descriptive analysis was used to summarize the morphometric measurement of length and weight as well condition factors of rescued pangolins and conducted the correlation and regression analysis [22]. Regression analysis and line parameters, a (intercept) and b (slope), made with log-transform statistical analysis were considered significant at 5% (*p* < 0.05). The coefficient of determination (r^2^) is a measure of the quality of the linear regression’s model prediction; a value close to 1 means a better model and correlation is significant at the 0.01 level (*p* < 0.01). Pearson correlations were analyzed at the *p* < 0.05 level of significance. In addition, Kendall’s tau-b and Spearman’s rho tests were used to analyze the corrrelation between length and weight. The correlation of bodyweight with Fulton’s condition factor (K_F_), relative condition factor (K_R_) and calculated weight (W_C_) were tested by correlation tests.

## 3. Results

### 3.1. Morphometric of Chinese Pangolin 

A total of 282 samples, where 59.22% of male and 40.78% of female rescued pangolins were accounted for in this study. The maximum total length and bodyweight of the species were 114.9 cm and 7566 g, respectively. The average ratio of total length and tail length of the Chinese pangolin was 1.7 (SD = 0.4). Pangolins older than one year (n = 123, 43.6%) were rescued and rehabilitated, followed by 6 month old pangolins (n = 82, 29.1%), while the adult pangolins were 2 years old (n = 42, 14.9%), and 3 years old and above (n = 35, 12.4%). The total length increase was statistically significant different between ages (F = 192.3, df = 3, *p* = 0.000) (Figure 1, Appendix A).

### 3.2. Length-Weight Relationships (LWRs) and Growth Type 

The correlation between total length (TL) and bodyweight (BW) of the Chinese pangolin (CP) was highly positive (around 80%). The bodyweight grows faster relative to increasing length; however, the growth type was negative allometric growth (b < 3) (Table 1). The regression model indicated that the power trend line was more fitting, and higher for males (r2 = 0.76) than females (r2 = 0.64) (Figure 2; Appendix A).

### 3.3. Condition Factors and Well-Beings 

Fulton condition factor (K_F_) values of the Chinese pangolin fluctuated between 0.23 and 2.12 whereas the average value of the relative condition factor (K_R_) was 1.04 ± 0.24 (Table 2). Nearly half of the overall samples (47.5%) of rescued pangolins were normal and 38.7% were in good health condition whereas a small percentage (13.8%) were in poor condition during the rescue (Table 3). Although the calculated weight of male sub-adults (1 year old) and adults (2 years old, and 3 years old and above) was higher, the relative condition was slightly higher in females (1.3 ± 0.1) which showed that females were fatter and healthier at this age than male individuals. However, the Fulton condition factors (F = 4.91, df = 3, *p* = 0.002), calculated weight (F = 190.42, df = 3, *p* = 0.000), and relative condition of ages were significantly different (F = 4.26, df = 3, *p* = 0.006) (Figure 3, Appendix A).

### 3.4. Monthly and Seasonal Variations in LWRs and K in Adults

The adult male and female pangolins measured between 75.2–103 cm and 66–114.9 cm from the snout to the tip of the tail and weighed between 4–7.6 kg and 3–5.8 kg, respectively. Adults also showed negative allometric growth (b < 3) as their length increased but positive allometric growth (b = 3.6) in June as the bodyweight was proportional with respect to length increase (Appendix A). The relative condition factor was greater than or equal to 1 (K_R_ ≥ 1) which showed that most rescued adults were in normal (59.7%) and good (39%) health conditions. However, the mean values of the Fulton condition factor (K_F_) and relative condition factor (K_R_) fluctuated in different months (Figure 4a) but K_R_ is statistically different between months (F = 12.887, df = 11, *p* = 0.000). The adult pangolins both male and female were found to be slightly fatter, and their wellbeing was higher in the wet season; however, K_F_ and K_R_ were not statistically significantly different between dry and wet seasons (*p* > 0.005) but they were positively correlated with each other (Figure 4b,c, Appendix A).

## 4. Discussion

The population of Chinese pangolin in Taiwan was comparatively increased due to conservation initiatives and strict implementation of rules and regulations after 1970 [23]. Many live pangolins have been rescued from threatening situations in Taiwan including across a range of countries every year. The rescued rate of live Chinese pangolins was extremely higher in southern Taiwan (30–40 individuals/year) in comparison to Nepal (4–5 individuals/year) and verified the increasing population in Taiwan [2]. They were released back into natural and safe habitats, such as protected areas, and conservation forests for their long-term conservation [24].

Male pangolins were found to be bigger and longer than females in different age groups, like in other mammalian species [25]. Body size affects almost every aspect of the biology of the species from physiology to life history to ecology [26]. The maximum length of Taiwanese pangolins was found to be longer than rescued pangolins from other countries, such as Nepal (Suwal, T.L. unpublished data) and China [8]. The maximum bodyweight was found more or less similar to a previous study in Taiwan [10], however, slightly less than Nepali pangolins, whereas slightly higher than in male pangolins from China (Table 4). The bodyweight and total length of adult rescued pangolins (2 years old and above) in different parts of the country as well as different countries were varied due to inhabiting the same species in different topography, geography, climate, and eco-regional zones along with the time [27].

This is the first length-weight relationships and condition factors study on Chinese pangolin. We used the allometric equation (W = aL^b^) to understand the LWRs which have been used widely to understand the biometric length-weight relationships in mammalian species [28,29,30,31,32,33] along with a study on tree pangolins (*Phataginus*
*tricuspis)* in Southwestern Nigeria [15]. The negative allometric growth deduced for nearly all analyzed pangolins (b < 3) suggested that this species has relatively fast growth in length. In addition, the high coefficient of determination values obtained in the assessment of LWRs over the whole year means there is good quality in the prediction of linear regression for the Chinese pangolin, suggesting that extrapolation in future confiscation can be conducted [34].

The majority of adult rescued pangolins (about 98%) were found in normal and good health condition. It could be due to sufficient food supply and minimized threats along with facilities and knowledge on handling of species during rescue and transformation in Taiwan. Therefore, veterinary training including species handling for the government authorities and stakeholders is of utmost need to minimize the unwanted effects on the species’ health for in-situ and ex-situ pangolin conservation. Female individuals of these ages; sub-adults (1-year old) and adults (2 years old and those 3 years and older) were slightly fatter and healthier than males. This might be due to sex-specific consequences of variation in fat reserves [35]. In female mammals, reproductive success is correlated with body condition; reproductive traits, such as litter mass, the number of litters, neonatal mass, and breeding lifespan increase with body condition [36,37]. However, the bodyweight and condition factors of rescued pangolins of both sexes in wet seasons seemed bigger and better than during dry seasons which showed that there were more survivability chances for species due to the availability of sufficient foods and active feeding behavior in wet seasons, with minimal threats. Additionally, pangolins needed to be digging more for termites due to the scarcity of ants, the primary source of food, in the dry season. It could be surmised that the paucity of food during the dry season causes a loss in bodyweight resulting in poor health conditions [38].

Understanding the well-being of live rescued animals is vital for species that are Critically Endangered, such as the Chinese pangolin. We recommend using the Fulton condition factor (K_F_) for the quick assessment of the health status of rescued individuals. This is because it can be calculated easily with a simple formula, and K_F_ values do not differ significantly across sexes and ages (Figure 3b). This study also suggested that individuals with K_F_ = 0.8 or >0.8 were identified as healthy individuals. Thus, the results can be used to determine whether the rescued Chinese pangolin is healthy enough for release, to develop rescue-rehabilitate-release guidelines, and design custom-made strategies to help protect this endangered species.

## 5. Conclusions

This study provided the first data on length-weight relationships (LWRs) and condition factors for the Chinese pangolin. Almost all LWRs showed a negative allometric growth of species. The Fulton condition factor (K_F_) and relative condition factor (K_R_) for the majority of rescued pangolins (around 90%) were close to 1 and greater and equal to 1, thus showing an overall state of wellbeing of the rescued pangolins. It is only recommended to release healthy animals with K-values of 0.8 or higher back into the wild in order to increase their survival rate. Similarly, K_F_ and K_R_ fluctuated in different months, but it is not statistically different between dry and wet seasons in adults. However, overall pangolins were found in better condition in the wet season compared to the dry season which might be due to the increase in feeding activity and food availability when temperatures are higher. Thus, this study fulfilled the aims set for it, and the data presented might constitute a valuable guideline for establishing future biometric studies for live pangolins in other range countries.

## Figures and Tables

**Figure 1 animals-12-00910-f001:**
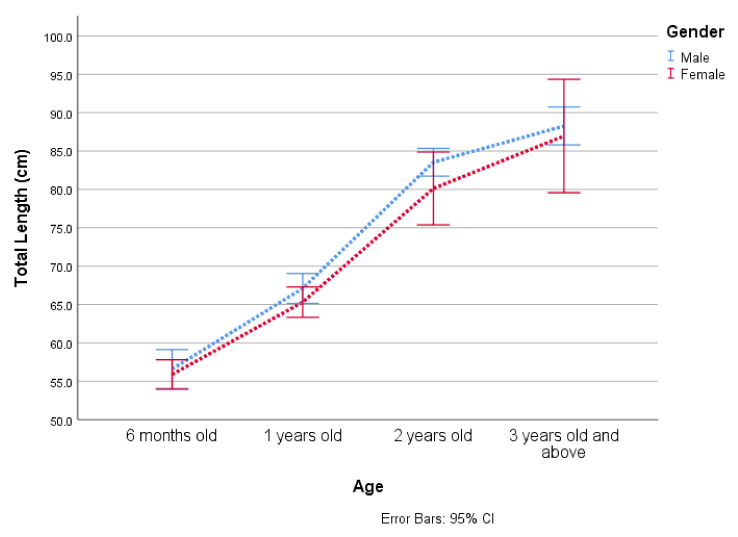
Age-wise total length growth of Chinese pangolins by gender.

**Figure 2 animals-12-00910-f002:**
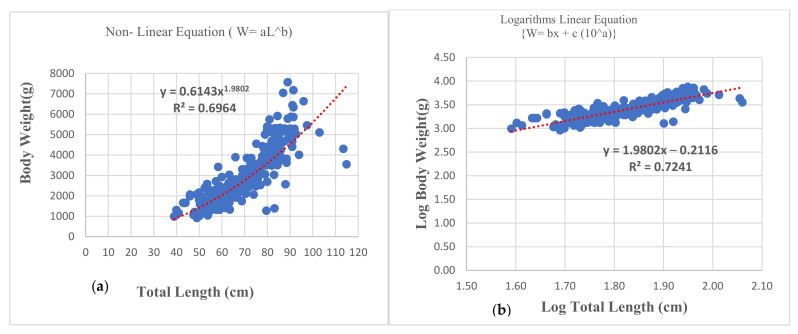
Total length-weight relationships of rescued Chinese pangolins (where a = Intercept and b = Slope of equation).

**Figure 3 animals-12-00910-f003:**
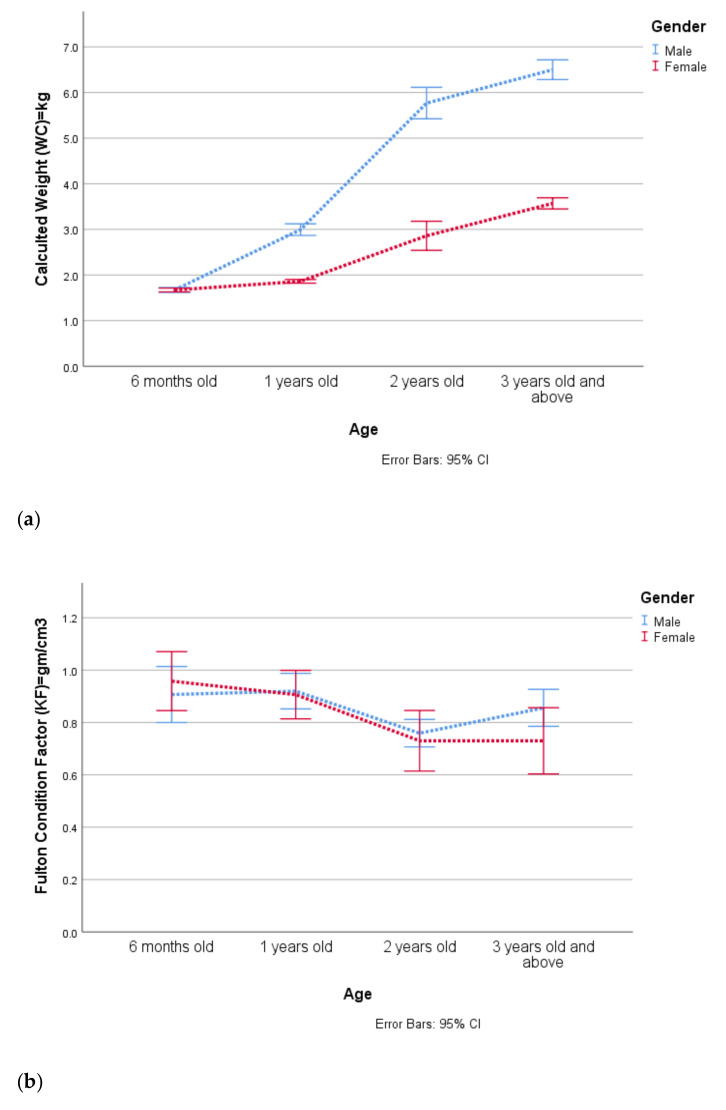
Age-wise (**a**) calculated weight and condition factors; (**b**) Fultion condition factor and (**c**) relative condition factor of Chinese pangolins.

**Figure 4 animals-12-00910-f004:**
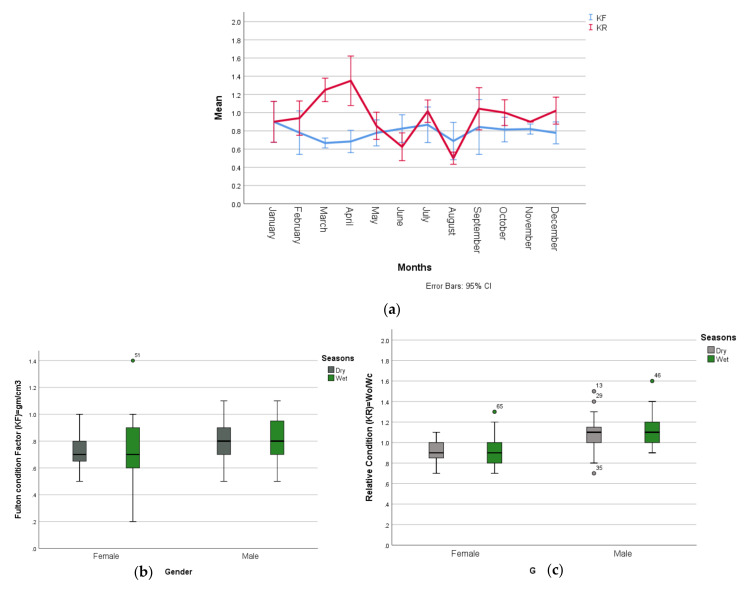
The variation of Fulton condition factor and realative condition factor in different (**a**) months and (**b**,**c**) seasons in adult Chinese pangolins.

**Table 1 animals-12-00910-t001:** Length-Weight Relationships {LWRs; ln (BW) = ln a + b ln (TL)} and growth type of different sexes.

Sexes	Logarithmic Transformation		Correlation Co-Efficient ‘r’	Non-Parametric Correlation ‘r’		Regression Co-Efficient ‘r’	Growth Type
	Intercept ‘a’	Slope ‘b’	PC	KC	SC		
Male	0.33	2.14	0.86 **	0.71 **	0.88 **	0.76	A-
Female	2.15	1.67	0.82 **	0.66 **	0.84 **	0.64	A-
Combined Sex	0.61	1.98	0.84 **	0.70 **	0.87 **	0.70	A-

PC: Pearson correlation, KC: Kendall’s tau-b, SC: Spearman’s rho, A-: Negative allometric,m, **: Correlation is significant at the 0.01 level (*p* < 0.01).

**Table 2 animals-12-00910-t002:** Condition factors of Chinese pangolin in different sex.

Condition Factors	Minimum	Maximum	Mean ± SD	CL_95%_	*t*-Test Sig.
K_F_					
Male	0.24	2.12	0.89 ± 0.28	−0.08–0.06	0.782
Female	0.23	2.08	0.88 ± 0.31		
Combined sex	0.23	2.12	0.88 ± 0.30		
W_C_					
Male	838.29	6698.62	3034.50 ± 1237.67	−491.24–11.55	0.061
Female	1056.24	5904.03	2425.36 ± 803.31		
Combined sex	862.26	7324.26	2721.14 ± 1058.65		
K_R_					
Male	0.33	1.73	1.00 ± 0.23	−0.13–0.01	0.018 **
Female	0.6	1.66	1.04 ± 0.22		
Combined sex	0.36	1.78	1.04 ± 0.24		

SD: Standard deviation, CL: Confidence limit for mean values, K_F_: Fulton’s condition factor, W_C_: Calculated weight, K_R_: Relative condition factor, *t*-test sig: *t*-test significant (*p* < 0.05). **: Correlation is significant at the 0.05 level.

**Table 3 animals-12-00910-t003:** Health status of rescued Chinese pangolin based on the Fulton condition (K_F_) and relative condition (K_R_) factors.

Health Status	K-Values	Fulton Condition (K_F_)		Relative Condition (K_R_)	
		N	%	N	%
Good	K > 1	46	16.3	109	38.7
Normal	K = 1 or near to 1	151	53.6	134	47.5
Poor	K < 0.8	85	30.1	39	13.8

N: number of samples; %: Percentage of samples.

**Table 4 animals-12-00910-t004:** The average total length (TL) and bodyweight (BW) and of adult (2 years old and above) Chinese pangolins.

Gender/Variables	Current Study		Previous Studies		
	2 Years Old, Taiwan	3 Years Old and Above, Taiwan	Suwal, T.L. (Unpublished Data), Nepal	Chin et al., 2015, Taiwan	Wu et al., 2004, China
Male					
TL	83.5 (75.2–94) cm, n = 22	88.3 (79–103) cm, n = 25	82.8 (48–94) cm, n = 19	x	74.9 (59.6–89.0) cm, n = 18
BW	4.4 (4.0–4.9) kg, n = 22	5.7 (5.0–7.6) kg, n = 25	4.2 (2.5–6.2) kg, n = 19	5 (3.5–7.6) kg, n = 19	4.5(2.1–8.5) kg, n = 20
Female					
TL	80.1 (66–114.9) cm, n = 20	87 (78–113.5) cm, n = 10	75.3 (57–93) cm, n = 13	x	69.9 (59.8–81.0) cm, n = 20
BW	3.5 (3–3.9) kg, n = 20	4.6 (4–5.8) kg, n = 10	3.5 (2.5–4.8) kg, n = 13	4.7 (4–6) kg, n = 14	3.5 (2.2–5.7) kg, n = 20

## Data Availability

The data of this study are available from the corresponding authors upon reasonable request.

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
