# Peer review of "Morphometric Relationships, Growth and Condition Factors of Critically Endangered Chinese Pangolin (Manis pentadactyla)"

_animals, 2022, doi:10.3390/ani12070910_

Round 1

Reviewer 1 Report

This paper is a study on the morphometric relationship, growth and condition factors of Chinese pangolin in Taiwan, it is very important for us to understand animal health condition and well-being. However this field research in Chinese pangolin is lack of work, thus this study is needed and significant. This study used 282 samples  of Chinese pangolin individuals, the sample size is sufficient which can be satisfied with the statistical need, ensure the reliability of the results. It is not easy to get such numerous critically endangered living Chinese pangolin. This study provide rich basic data on Chinese pangolin's biology. The authors know well relevant research background, research methods are feasible, data process is correct, results are reliable, conclusion is credible, discuss is reasonable. I think the manuscript reach the demand to publish, after revision it can be considered to release in the journal Animals. Only one suggestion to authors for revision that is adult and non-adult Chinese pangolin how to identify? Others marked the manuscript.

Author Response

Respected Reviewer,

Reviewer 2 Report

This is a very well written paper examining morphometric parameters of a CE mammal, the Chinese pangolin. Similar data are important for understanding the well-being of the species and trace its conservation. The paper aims to successfully contribute to this end.

The authors use a large dataset of pangolins and the results are statistically robust. However, although the authors claim that the goals of the study were fulfilled it is necessary to involve the conservation parameter. This should be clearly put among the aims of the paper (e.g., how can these data be used to infer on the conservation status of the species in Taiwan and worldwide) and thoroughly discussed afterwards.

REVIEW OF animals-1652932 This is a very well written paper examining morphometric parameters of a critically endangered mammal, the Chinese pangolin. The topic is original because it is the first time, where similar data have been calculated for the species at this extent. In effect, the presented data represent a large dataset and the standardized methodology used assures that they are statistically robust. Similar data are very important for understanding the well-being of the species and when compared with previous analogous databases provide evidence for common patterns in growth that can be related to the ecology, energetic demands, and sex differences of the species. The results are very clear and well-presented, and the discussion flows very well and is based on the findings of the study and the available comparative works. There is only one drawback. Although the authors claim that the goals of the study were fulfilled, it is important for them to involve the conservation parameter. Similar data are important for understanding the well-being of animals and are thus vital for species that are Critically Endangered as the Chinese pangolin. This perspective should be clearly put forth among the aims of the paper in the introduction section (e.g., how can these data be used to infer on the conservation status of the species in Taiwan and worldwide) and then of course thoroughly discussed in the discussion and conclusion section (e.g., these data help distinguish healthy from non-healthy animals and help design custom-made strategies for protecting the species).

Author Response

Respected Reviewer,
